# Reclaiming motherhood through shame, distance, and gratitude—A phenomenological study of Swedish women's lived experiences of giving birth while ill with COVID-19

**Maria Revelj**[1,2]*, **Anna Wessberg**[2,3], **Ylva Carlsson**[2,4], **Maria Lindqvist**[5,6], **Verena Sengpiel**[1,2], **Karolina Linden**[3]

1 Department of Obstetrics and Gynecology, Institute for clinical sciences, Sahlgrenska Academy, University of Gothenburg, Gothenburg, Sweden, 2 Department of Obstetrics and Gynecology, Sahlgrenska University Hospital, Gothenburg, Sweden, 3 Institute of Health and Care Sciences, Sahlgrenska Academy, University of Gothenburg, Gothenburg, Sweden, 4 Centre of Perinatal Medicine & Health, Institute of Clinical Sciences, Sahlgrenska Academy, University of Gothenburg, Gothenburg, Sweden, 5 Department of Nursing, Umeå University, Umeå, Sweden, 6 Department of Clinical Sciences, Obstetrics and Gynecology, Umeå University, Umeå, Sweden

* Maria.Revelj@gu.se

## Abstract

### Introduction

Pregnant women were one of the most exposed and vulnerable groups during the COVID-19 pandemic. While much is known about the general effects of the pandemic on pregnant women's well-being, little research has focused on the experiences of women who gave birth while infected with SARS-CoV-2.

The aim of this study was to gain a deeper understanding of the lived experiences of women who gave birth while being ill with COVID-19.

### Materials and methods

This is a qualitative study utilising a phenomenological reflective lifeworld approach to explore the lived experiences of Swedish women (n = 10) who gave birth while ill with COVID-19 between April 2020 and May 2021.

### Results

The essence of the women's experiences was described as 'Reclaiming motherhood through shame, distance, and gratitude,' supported by four constituents: "feeling intense shame and guilt for getting infected"," striving to overcome distance in the birth setting", "experiencing gratitude for receiving compassionate care" and "trying to comprehend motherhood and fighting to be reunited".

**Data availability statement:** Data Availability: Data contains potentially identifying and sensitive information, and there are ethical restrictions to sharing the data file as it includes individual interviews with participants. For data requests, please contact The Swedish Ethical Review Authority: registrator@etikprovning.se.

**Funding:** The author(s) received no specific funding for this work.

**Competing interests:** The authors have declared that no competing interests exist.

## Conclusion

A nuanced understanding of the experience at the intersection between childbirth, illness and the societal context is imperative for healthcare professionals and society to provide optimal care for one of the most vulnerable groups during pandemics, pregnant women and their newborns.

## Introduction

Giving birth during the COVID-19 pandemic was an extraordinary experience owing to the uncertainties and isolation that came with this global health crisis. Pregnant women and their partners' psychological well-being was negatively influenced by changes in maternity care implemented due to the pandemic [1–3]. Although parental experiences of pregnancy and birth during this unique time have been well studied [1], very little is known about the experiences of women infected with SARS-Cov-2 at the time of giving birth. A French qualitative study that included pregnant women with mild COVID-19 symptoms found that the idea that their partner might not be allowed to attend childbirth was experienced as difficult or intolerable. The study did, however, not focus on the experience of childbirth while being ill with COVID-19 [4].

More is known about how the pandemic affected pregnant women in general. A Swedish qualitative study on pregnant women and their experiences of uncertainties posed by the COVID-19 pandemic during pregnancy found that women experienced the pandemic as pervading all aspects of pregnancy, and that they had to deal with the new situation of the pandemic on top of all the thoughts and feelings that normally occur during pregnancy [2]. Fear and anxiety were common among pregnant women during the COVID-19 pandemic, as there were many unknowns [3,5].

A continuous learning process about the virus, its effects, and the specific factors associated with it took place throughout the pandemic. This meant that recommendations, guidelines, and healthcare policy changed rapidly, making it very difficult for women and their partners to prepare for labour and birth. Guidelines regarding personal protective equipment (PPE), the use of nitrous oxide for pain relief during labour, or even rules about being separated from one's newborn child directly after birth as a safety precaution were updated several times [2]. In addition, two recently published studies showed that hospital policy excluding an accompanying partner from the postnatal ward during the pandemic negatively impacted partners' satisfaction with care [6]. Partners want to be included in pregnancy and birth as preparation for their upcoming parenthood [7].

Childbirth and becoming a parent are life-altering events that potentially affect the woman throughout the rest of her life [8]. Giving birth during a pandemic, while infected with a novel virus with unknown risks both for herself and her unborn child, can be expected to impact a woman in multiple ways. Thus, this qualitative interview study aimed to gain a deeper understanding of the lived experiences of women who gave birth while ill with COVID-19.

## Materials and methods

### Study design

This qualitative study employed a phenomenological reflective lifeworld approach, as described by Dahlberg et al. [9], to illuminate the lived experience of giving birth while being ill with COVID-19. Phenomenology focuses on understanding and exploring the subjective experiences of individuals. The approach was initially developed by philosopher Edmund Husserl and later expanded upon by philosophers such as Martin Heidegger and Jean-Paul Sartre. Phenomenology is used in various fields of science and qualitative research in general. It involves gathering in-depth descriptions of people's experiences through methods such as interviews. The goal is to illuminate meanings and a possible essence of the phenomenon, described through its constituents, rather than to develop categories or themes. In the lifeworld tradition, one does not 'set aside' pre-understandings; instead, following Dahlberg et al., one bridles their pre-understanding by deliberately holding back premature interpretations [9]. The phenomenological approach was deemed most suitable for gaining a deeper understanding of the phenomena of being ill with COVID-19 while at the same time giving birth and becoming a parent. Women with different backgrounds were interviewed to capture the 'lived experiences' of the individuals and to identify the different nuances.

The study is part of the Swedish multicenter COVID-19 during pregnancy and early childhood study (COPE, trial registration number: NCT04433364) [10].

### Study context

In Sweden, pregnant women typically follow a free of charge standardised health care programme at their antenatal care clinic. While the programme is not mandatory, almost all pregnant women still choose to participate [11,12]. Sweden is one of the countries with the lowest maternal and perinatal mortality in the world [13]. Pregnancy and childbirth are typically a shared experience, and partners usually accompany the woman to antenatal care clinic visits and are present during childbirth.

The coronavirus SARS-CoV-2 originated in China and spread worldwide, beginning in late 2019. The World Health Organization (WHO) declared it a pandemic on the 11th of March 2020 [14]. At the time, it was challenging to fully comprehend the extensive consequences the virus and pandemic would have on the average person. The pandemic was going to affect virtually all aspects of life. Unlike many other countries, Sweden did not have an official lockdown, and the incidence of COVID-19 was high. Several recommendations were introduced in society to reduce the spread of the coronavirus, such as social distancing and working from home when possible [15]. Pregnant women were early on identified as risk group for severe illness [15]. SARS-CoV-2 infection during pregnancy was shown to be associated with preterm birth [16], and cases of vertical transmission became known [16]. However, pregnant women were not considered a risk group by Swedish health authorities until February 2021 [17].

The healthcare system faced a massive challenge in handling all acute cases while maintaining regular healthcare services. Healthcare professionals had to care for people without knowing how to fully manage the situation [18]. Due to the uncertainty regarding the potential threat SARS-CoV-2 posed and to minimise the spread of the virus, most Swedish antenatal clinics restricted visits and did not allow anyone to accompany the pregnant woman to her appointments [3].

### Participants

Participants were recruited in September 2021 via an open hospital Instagram account as well as by clinical staff. Interested women were asked to contact the interviewer via email to receive oral and written information about the study. All women aged 18 or older, who spoke Swedish or English and gave birth in Sweden while having active COVID-19, were eligible for the study. Eleven women were recruited; however, only ten women were eligible for the study (one did not have COVID-19 at the time of giving birth). The women who participated gave birth at five different hospitals located in three of

Sweden's 21 independent healthcare regions. The sample size was determined in line with phenomenological methodology, which prioritises depth of experience over breadth [9]. Participants represented a variety of socioeconomic, obstetric and health-related backgrounds, including both primiparas and multiparas, and women with and without pregnancy complications.

Prior to the start of the interview, women were reminded that their participation was voluntary, and that they could at any time, without providing an explanation, withdraw from the study. The women gave their written consent before the interview. The women had given birth between April 2020 and May 2021. Interviews were held between October and December 2021, 5–18 months after birth. All interviews were conducted in Swedish. The interview was carried out at the participant´s preferred time. All interviews were conducted by the same interviewer (MR) who was not involved in the clinical care of any of the participants.The interviews were conducted digitally via video meetings, meaning that the participant was comfortable in her own home setting. The interviewer presented herself as a PhD-student, not as an obstetrician. She also used her own casual clothing, mostly hoodies or fleece jackets, and the background was neutral with no markings of a hospital setting to mitigate any potential power imbalance in the interview setting. Interviews started with an open-ended question: "Please tell me about your experience of giving birth while ill with COVID-19?" Follow-up questions, such as "Can you elaborate on that?" "How did that make you feel?" "How do you mean?" were asked to deepen the understanding. The interviews lasted between 30 and 54 minutes, (mean = 40 minutes).

## Data analysis

The data were analysed based on the phenomenological reflective lifeworld approach described by Dahlberg et al [9]. The interviews were recorded and transcribed verbatim. Two of the authors (MR & KL) read the interviews several times. Repeatedly reading the transcripts helped to gain an understanding and familiarity with the data. The second step was to find meaning units that corresponded to the aim of the study. The meaning units were then grouped into clusters. NVivo Windows Release 1/NVivo12 Mac software was used for extracting meaning units and grouping clusters. The final step involved moving between the clusters and the transcribed interview text to identify the essence. The essence was described by four constituents [9].

When the core essence emerged, it was meticulously explored, delving into its constituents and their various intricacies and subtleties. The authors needed to continually return to the text to ensure that the essence was truly built from the various clusters and make sure that the constituents formed a whole. The tentative essence was continuously challenged against variations and contrasts in the data, and by shifting focus between different accounts and perspectives we tested whether the structure held across these differences. This process, carried out in a bridled stance, ensured that the essence grew from the material rather than from the researchers' pre-understandings. Describing the essence required this constant oscillation between the text as a whole and its smallest parts. The first analysis was conducted by MR and KL and then discussed among all the authors to confirm the essence and its constituents.

Ethical approval for the study was obtained from the Swedish Ethical Review Authority (2020–02189 and amendments 2020–02848, 2020–05016).

## Results

Ten women participated in the study with a median age of 32 years (range 28−40). Half of the women gave birth at a university hospital while the other half utilized a regional or local hospital. All but one of the women were born in Sweden; one was born in another European country. Four women held a high school degree and six had gone to university. All women were married or cohabiting with their partner. Four women were primiparas, and four were diagnosed with a pregnancy complication (preeclampsia, HELLP syndrome, gestational hypertension, or cholestasis during pregnancy). The majority of women (n = 7) had mild to moderate COVID-19 symptoms, whereas three women were critically ill and required intensive care. Three of the children were born prematurely via caesarean section due to their mothers being severely ill with

COVID-19. There was an additional caesarean section in the group due to distressing fetal monitoring. Of the six vaginal births, three women were induced—two due to COVID-19 and one due to prolonged pregnancy. Three of the children were positive for SARS-CoV-2 at birth, and one became severely ill with COVID-19. One child had complications related to prematurity; the rest of the children were born healthy.

The essence of the phenomenon of women's experiences of giving birth while being ill with COVID-19 can be described as 'Reclaiming motherhood through shame, distance, and gratitude'. This experience was shaped by a profound emotional journey, where women navigated societal expectations, illness, and identity transformation while becoming mothers.

The pandemic was an extraordinary event that placed demands on the entire society as well as the individual. The women expressed deep shame for becoming infected with the virus. Many pregnant women took significant personal responsibility not only to avoid getting infected but also to prevent infecting others. This sense of failure contributed to feelings of guilt and shame, despite rational knowledge that they were not personally at fault. Media and healthcare facilities urged everyone to be cautious. To then actually catch the virus provoked feelings of guilt. Once it became clear that they were infected, the women wondered how their birth would proceed and whether anyone would care for them.

Being pregnant and giving birth are transformative events. Being ill with COVID-19 added a profound sense of physical, emotional, and communicative distance to the birth experience. Physical distance was expressed through isolation protocols and protective gear; emotional distance emerged as a lack of intimate contact or spontaneous connection; and communicative distance was shaped by the barriers posed by PPE and limited facial expressions. Therefore the distance emerged as physical, emotional and communicative, which all impacted the birth experience.

Despite the emotional and physical challenges of labour while ill with COVID-19, the birthing women felt immense gratitude for receiving care. Being cared for, treated, and seen as individuals, despite the risk of contagion, evoked deep feelings of gratitude for being treated as women giving birth, not as infection hazards. Those who were critically ill also harboured fears about the severity of their illness and feared for their own and their unborn child's health and life. Being separated from their child and family made it difficult to comprehend motherhood. A great fighting spirit emerged to recover and be reunited with their child.

The essence can be further described in four constituents: "feeling intense shame and guilt for becoming infected," "striving to overcome distance in the birth setting," "experiencing gratitude for receiving compassionate care," and "trying to comprehend motherhood and fighting to be reunited."

## Feeling intense shame and guilt for getting infected

The women described deep feelings of shame and guilt for falling ill. The pandemic was described as a time when they had to adapt to a new reality. Being pregnant was a new experience for some of them, while at the same time, the world became a different place due to the coronavirus. The way they interacted with others was defined by guidelines set by authorities to mitigate the spread of the virus and lessen the healthcare burden. Reporting on COVID-19 and its consequences for society, along with the ever-changing infection control measures, was constantly present in the women's minds. The feelings of shame and guilt for becoming infected were tangible, especially if they had breached or stretched the boundaries recommended by authorities.

> *Almost shameful. You feel like, "Oh my God, why… did I have to meet someone during Easter? Did I have to tempt fate? I know how contagious it is, I work with this..." – Participant 6, 0-para*

Some women felt intense shame, almost as if they had committed a criminal act, by falling ill. At the hospital, they were acutely aware that they were infected when they had to move through hallways or other shared spaces, fearing they might encounter others and potentially spread the infection. Not only did the women harbour the shame of becoming ill

themselves, but also the guilt of potentially infecting someone else and causing that person's death. The women explained that, although they could rationally understand they were not responsible for the pandemic and its consequences, the feeling remained that they could be a healthcare burden. The shame of possibly infecting others and potentially spreading the virus to loved ones, as well as to healthcare professionals who were helping others, weighed heavily on their emotions.

*"Yes, but you almost feel a bit like a criminal. It's such a strong... I mean, dangerous virus that's spreading, and then you bring it into a hospital while being heavily pregnant... I didn't want to infect anyone else." – Participant 8, 1-para*

She continued:

*"[...] When I walked down the corridor to leave... The only thing I kept thinking was, 'I hope no one comes. I hope no one comes.' It was more about... The fear of possibly infecting someone else who might not survive it, and then maybe finding out that this person actually died because of me." – Participant 8, 1-para*

In addition, several women felt that they were a burden. They felt that it was troublesome for the healthcare professionals with all the extra measures they had to take on their behalf.

*"And then you end up not calling for help unless absolutely necessary, knowing they have to put on all that protective gear, even if you need something. So, it felt... But once I had the confirmed COVID-19 diagnosis, it was a bit easier... You just didn't want them to catch it." – Participant 7, 1-para*

### Striving to overcome distance in the birth setting

Some women wondered how their births would proceed due to having COVID-19 and the precautionary measures that staff had to take to avoid infection. They reflected on how they would establish a connection with the midwife through the PPE. They also questioned whether someone would assist them and, if so, in what way.

Upon arrival at the hospital during labour, several challenges emerged due to the requirement for midwives to wear PPE. This was particularly evident to women who had experienced childbirth before the pandemic. The requirement to wear a face mask also applied to the birthing woman. Conveying emotions and needs through these barriers proved challenging.

*"Partly, I felt that I didn't get the same connection with the people in the room, maybe because I couldn't see their faces as well as during my first birth. However, the care was so good that I still felt very well taken care of and very safe with the staff present. But the lack of personal connection, which I had experienced during my first birth, was noticeable. There was a distance between us, likely because we couldn't see each other as clearly." – Participant 2, 1-para*

Women described the distance that emerged between themselves and the midwives as both physical and emotional. The physical distance was described as though the healthcare professionals were almost encased.

This physical distance contributed to the perceived emotional distance. One woman described it as a difficult barrier to overcome—a sense of detachment between herself and the midwife during childbirth.

*"In a way. It was almost like, 'okay, here I come,' when I walked down the corridor and saw them standing there in their protective gear. It felt more like, 'okay, we'll be here to catch your baby, but the rest you'll have to handle on your own.'" – Participant 7, 1-para*

The same woman described a desire for contact, expressing the need to feel closeness with the midwife assisting her, and longing for support.

> "...and I'm sitting in a chair and they're going to take the COVID test from my nose and throat. And then it felt like they were very far away. I don't know how far... or if it was five centimeters away, I don't know, but it felt very far away. And then I remember asking for 'hold on to me,' 'stay here,' 'be here,' or... I don't know if I said, 'stay here,' but I wanted someone to hold my hand." – Participant 7, 1-para

Some women, however, shared positive experiences where compassionate gestures broke through these barriers, reaffirming the human connection despite isolation.

> "Despite it all there was body contact, they put their hand on me and massaged me, and it still felt warm even though there was this sterile environment with clothes and all. And I think it would have felt that it was more difficult if I felt like they didn't want to be close, because I was infected and might have been dangerous in some way. But it still felt very much like 'we need to protect ourselves,' but otherwise, everything felt normal to me." – Participant 2, 1-para

Women felt that both the healthcare professionals and they themselves tried to navigate and find strategies to overcome the obstacles posed by PPE. Strategies to overcome the physical barriers associated with PPE also helped to reduce the emotional distance. One woman described how she attempted to learn more about the person behind the mask. This was an effort to make the situation less overwhelming and to humanise the person behind the mask.

> "I like to talk, so I chat with everyone. That was a bit of my salvation. I started asking questions like, 'Who are you? What do you do? […]' Their lives. Sure, it's their job, but it felt like much more than just a job, actually. The person behind the roles." – Participant 9, 3-para

### Experiencing gratitude for receiving compassionate care

The women expressed feeling grateful for receiving excellent care and treatment despite having COVID-19. This gratitude was, to some extent, intertwined with the feelings of shame and guilt that also pervaded their thoughts, creating an ambivalent emotional landscape.

> "The entire interaction was very professional and kind, even though they were wearing those space suits, and that was something they joked about as well." – Participant 1, 1-para

The women described feeling grateful for healthcare professionals who, despite the risk of infection and the potentially deadly consequences of COVID-19, dared to provide care in a supportive and non-judgemental manner. The women were especially grateful for being seen as individuals and as birthing women, not merely as carriers of the virus. They felt that the healthcare professionals did their very best in a challenging situation.

> "But you also understand that they're doing their absolute best, and it was really tough for them to haul the CTG machines back and forth, in and out with their protective clothing. So, you have a lot of appreciation for how they made the best of the situation, truly. And it felt like they really did." – Participant 3, 0-para

Women who had been critically ill particularly described feeling grateful toward the healthcare professionals. They were especially thankful that the healthcare system functioned so well despite this extraordinary situation.

*"Yeah. But besides highlighting the care... the care has been fantastic. We talked a lot about it when... if you have a pain in your shoulder, for example, it's a bit of a hassle in the healthcare system and you go to the health centre and you're always dissatisfied. But I was incredibly impressed with the care. How it worked when you're in a situation between life and death. I'm extremely, extremely grateful and think the staff is amazing. Quick decisions, quick actions, and... yeah."* – Participant 9, 3-para

**Trying to comprehend motherhood and fighting to be reunited**

The women who were critically ill were separated from their children after birth. The separation and lost opportunity for immediate bonding impacted their experiences of early motherhood and affected them deeply. Becoming a parent while fighting for one's survival was portrayed as a multifaceted and painful experience. For some women, it was difficult to comprehend who the child was and that they had become a parent. They experienced a sense of unreality, feeling as though they were in a twilight zone: Had the birth really happened?

*"I hadn't even realized that I had had a baby. So, he [the baby], ended up in the neonatal intensive care unit… up there, and I ended up in the ICU and had to fight for my life, simply put."* – Participant 9, 3-para

Another woman had to ask the healthcare professionals to pause so that she could see her child before they were separated.

*"But otherwise, she was alert. So, I said, 'But stay, I just need to see her.' So I touched her hand for maybe five seconds, and then they took her away."* – Participant 6, 0-para

The women described being entirely dependent on others who acted as intermediaries between them as parents and their child. They had to allow healthcare professionals and relatives to look after their child when they were unable to do so themselves. This was an emotional and mental strain. One woman described how she did not even recognise her child when they met for the first time.

*"When my husband came to get me with the wheelchair, he rolled me toward our room, and there was a nurse feeding a baby in front of me. It was outside the room, not in our room. I didn't understand at first because she was looking at me a lot, and I was looking at her. Then, after we had rolled a bit further, I just said, 'Wait a minute, is that my baby?' 'Yes, that's your baby.'"* – Participant 5, 1-para

Missing the experiences of early motherhood filled the women with sadness. Several women described sorrow over the birth and becoming a mother not turning out the way they had anticipated or imagined.

*"But it was the feeling that I was just a body and not a new mother, and it was a sorrow to miss out on… to miss the whole experience as well."* – Participant 5, 1-para

Another woman also described the great fear that lingered in the back of her mind. Would she survive and get to go home? What would happen to her children if she died? She was confronted with the realisation that the situation was very serious.

*"One Saturday when my husband came, I became really scared. That's when I felt it was serious. I remember looking at him and just saying, 'We were supposed to grow old together.' […] And he just said, 'Yes, we were,' and then added,*

*'Of course we are.' So he corrected himself, you know. But I think he and I both understood the gravity of the situation there. […] The thought of losing him and… 'My family and the boys, what will happen to them?' It became so overwhelming… " – Participant 9, 3-para*

For some women, a profound existential desire to recover, to reclaim their life, and to embrace their newly gained motherhood emerged. They exhibited an immense determination to overcome the challenges of reuniting with their child.

*"No, but you can't stand here talking about food," I said. "Someone just died in the room next door. You work here, you can handle this. I can't handle this." Then I looked at the pregnant woman and said, "She's getting worse, I'm getting better; we're sharing a room. My story doesn't need to become her story. Someone is dying in the room next door. I need to go home now." And then it was almost… yeah, but 45 minutes later, I was sitting in a transport home. – Participant 9, 3-para*

Another woman described how the desire to be together, to become a family for the first time, became a powerful driving force, overriding her concern for her own body and health. For her, the need to be reunited with her family outweighed the recommendations to remain at the hospital as advised by her medical team.

*"…just to be able to go home and sleep together [as a family], against all recommendations. And then the next day we were back and … Yeah, I was oxygenating myself … I was down to 80[%] when I took a little walk around the room and thought, 'Oh, it'll be fine, I'll manage. I'll just live on the couch and my partner can help me with everything.' So, I actually discharged myself the … next day, because I felt I needed to be with my daughter and my family." – Participant 6, 0-para*

## Discussion

This study provides in-depth insight into women's lived experiences of giving birth while ill with COVID-19. Giving birth while being ill is a vulnerable situation, but being afflicted with a disease regarded by society as a significant public health threat evoked thoughts and emotions that manifested among these women as a mix of shame, guilt, and gratitude while they fought to reclaim motherhood. The intensity of their thoughts on how to relate to the unknown varied between the women.

The sense of guilt and shame from contracting a disease deemed dangerous by society was profound. The participants understood that feeling shame was not rational, yet the emotion persisted. Shame was closely linked to guilt and fear, adding an additional psychological burden beyond the infection. Social stigma following infection has been reported in a German mixed-method study of individuals who recovered from COVID-19. This stigma was present regardless of whether the person experienced mild or severe symptoms [19]. Various nuances of social stigma became evident in the interviews. A sense of responsibility towards those around them, including relatives and healthcare workers, was expressed. The emotional burden of not only caring for oneself in a vulnerable situation but also for others was significant. Healthcare professionals need to be aware of these feelings and endeavour to mitigate them, as these emotions might influence postnatal mental health and potentially affect the parent-child bond [20]. Simple measures such as explicitly reassuring women that infection is not their personal failure, and acknowledging the stigma they may feel, might reduce the weight of guilt and shame.

Women expressed uncertainty about how they would be cared for during labour and birth. This uncertainty was expressed both in terms of practical matters, such as where they would physically give birth, and emotional concerns, such as whether they and their partner would be welcomed at the birth unit and well cared for. These emotions varied in intensity. Encountering healthcare professionals wearing full PPE, who had to perform additional tasks just to assist,

was sometimes a barrier to seeking help, even when it was needed. While there was an understanding of the necessity for PPE among healthcare professionals, there was also a concurrent concern about the ability to connect emotionally through the protective barrier. This was similarly identified as a barrier by midwives assisting birthing women in full PPE in a Swedish qualitative study by Göransson et al. Gradually, as the pandemic evolved, midwives adopted strategies to build an emotional connection with women despite the obstacles posed by PPE [21]. During the pandemic, some researchers suggested that healthcare professionals could attach their portraits to their PPE in an effort to humanise care [22,23]. However, this approach was not experienced by the women in this study. Our findings underscore the importance of humanising care when protective measures are required; even small gestures, such as verbal affirmation, intentional eye contact, or physical touch when safe, might help overcome the barriers imposed by PPE.

Gratitude was evident throughout all the interviews, particularly among those women who were severely ill. Women were grateful not only for the care they received but also for being recognised as individuals and as birthing mothers, rather than merely as carriers of the virus. This sentiment was reflected by midwives in the previously mentioned qualitative study by Göransson et al. Midwives made every effort to support women in labour and facilitate a normal birthing process, despite wearing PPE, the risk of contagion, and COVID-19-related restrictions [21]. This profound gratitude has also been described by COVID-19 survivors who were treated in the ICU, underscoring the importance of holistic care in life-threatening situations [24,25]. Maintaining a focus on the woman as a birthing mother rather than an infection risk is a central practice pointer, reminding professionals that respectful, compassionate presence is therapeutic in itself.

The most severely ill mothers expressed a profound desire to reunite with their child and a willingness to risk their own health to be together. This highlights the complexity of understanding motherhood when one has scarcely or not at all seen one's child at birth, and when the immediate post-birth focus is on one's own survival. The shift of focus from one-self to one's child, and the distress of missing the precious early moments with one's child, was previously described in a qualitative study of women who nearly died during pregnancy and childbirth [26]. To date, there are no studies specifically following up on women with COVID-19-related near-death experiences or exploring how healthcare professionals can best support these mothers. Previous research has shown that traumatic childbirth experiences can be linked to post-traumatic stress disorder and postpartum depression, with potential long-term consequences for maternal mental health and bonding with the child [27,28]. One way to prevent the development of post-traumatic stress disorder and postpartum depression is to practise trauma informed care. Rather simple care interventions such as encouraging skin-to-skin care as soon as possible, providing education concerning the natural course of stress when undergoing trauma, and encouraging expressive writing about one's experiences can empower the woman following a traumatic experience. These simple approached could prove very useful when caring for women severely ill during pregnancy and birth. Further, these women could benefit from a follow-up appointment with a psychologist trained in trauma informed care [29]. Clearer clinical protocols to support early skin-to-skin contact when possible, and systematic psychological follow-up for women who have been critically ill, are important lessons for crisis preparedness.

## Strengths and limitations

Using the phenomenological lifeworld approach is a strength of this study as it allows the researcher to illuminate the individual's perspective, providing rich, detailed descriptions that capture the essence of how individuals perceive and engage with the phenomena [9]. The phenomena studied exist within a specific context involving women receiving obstetric care in Sweden, but the findings and conclusions align with previous studies on women who were critically ill during childbirth [26], and non-pregnant COVID-19 survivors [19,24]. Phenomenology seeks to understand and describe the subjective experience of a phenomenon. To strengthen trustworthiness, we sought to describe the women's lived experiences with openness, nuance, and methodological transparency. With respect to credibility, we have presented rich empirical excerpts to illustrate our interpretations. For confirmability, the identified essence and constituents were critically discussed and validated with co-authors who were not involved in the primary analysis. Regarding

dependability, we ensured transparency by carefully documenting the analytic process, moving iteratively between whole and parts, and by engaging in reflexive discussions about our pre-understandings throughout the study. Our findings might be transferable to other settings; however, contextual factors must always be considered when applying findings to new environments. We also acknowledge that the part of the recruitment that took place via the hospital's Instagram account may have inadvertently excluded women with lower levels of digital literacy, which could in turn affect the transferability of the findings.

To prevent bias due to the researchers' preconceived notions, the first author (MR), who also conducted the interviews, engaged in ongoing reflexivity throughout the research process. This involved reflecting on personal background and pre-existing assumptions that could influence data collection and analysis. While phenomenological research within the lifeworld tradition does not apply bracketing or epoché in the Husserlian sense, we used 'bridling', deliberately holding back pre-understandings and deferring premature interpretations [9]. By employing the technique of 'bridling', the author consciously delayed immediate interpretation of the data, thereby prioritising the participants' voices. This approach facilitated an open-minded exploration of the participants' experiences, aiming to keep the findings grounded in the women's perspectives. Furthermore, all the analysis was thoroughly discussed with the last author (KL), who also delved into the data to gain a deeper understanding of the participants' experiences. MR, YC and VS are obstetricians with clinical experience of caring for severely ill pregnant women with COVID-19. AW, ML and KL are midwives. While AW cared for birthing women during the pandemic, ML and KL did not conduct clinical work during the pandemic. YC, VS, ML and KL conducted several studies regarding pregnancy and COVID-19.

Efforts were made to use a non-directive interview technique, beginning with open-ended questions and allowing participants to steer the conversation. This approach minimised the risk of leading the participants and ensured that their authentic lived experiences were captured, rather than being shaped by the researcher's expectations. Recognising the unavoidable positionality associated with the first author's (MR) institutional affiliation in obstetrics, she underscored confidentiality and her independence from clinical services, clarified that participation would not affect care, and emphasised an open, bridled stance in the interviews. This stance included encouraging participants to describe their experiences in their own words and remaining attentive to nuances, thereby allowing the phenomenon to show itself rather than being shaped by the researcher's assumptions.

Extensive efforts were made to ensure methodological transparency, detailing the process of working with the text and how the analysis and essence emerged from the interviews. The essence of the phenomena and the selected constituents were continually discussed among the researchers. The analysis was carried out entirely in Swedish to stay close to the participants' original expressions. For publication purposes, selected quotations were translated into English, and these translations were reviewed by several authors to safeguard the preservation of nuances. The study is reported according to the Standards for Reporting Qualitative Research (SRQR) [30] to ensure methodological transparency.

## Conclusion

In conclusion, this qualitative phenomenological study reveals the intricate tapestry of emotions and challenges experienced by women giving birth while ill with COVID-19. The essence of the phenomena can be described as 'Reclaiming motherhood through shame, distance and, gratitude' supported by the constituents of "feeling intense shame and guilt for getting infected"," striving to overcome distance in the birth setting", "experiencing gratitude for receiving compassionate care" and "trying to comprehend motherhood and fighting to be reunited". A nuanced understanding of the experience at the intersection between childbirth, illness and the societal context is imperative for healthcare professionals and society to provide optimal care for one of the most vulnerable groups during pandemics, pregnant women and their newborns. This study offers insights to inform crisis preparedness in maternal healthcare. It highlights the need for policies that safeguard not only physical safety but also women's dignity, and emotional well-being. Maintaining psychosocial support and respectful care remains essential, even during public health emergencies.

## Key message

Understanding the experiences of women who gave birth while ill with COVID-19 is crucial for optimizing healthcare. Highlighting the need for compassionate, individualized support during pandemics.

## Supporting information

**S1 File. SRQR checklist.** Completed checklist based on the Standards for Reporting Qualitative Research (SRQR) guidelines.
(DOCX)

## Acknowledgments

This study is part of the "COPE (COVID-19 during pregnancy and Early childhood) study", NCT04433364. We would like to thank the women who participated in the study.

## Author contributions

**Conceptualization:** Anna Wessberg, Ylva Carlsson, Verena Sengpiel, Karolina Linden.

**Data curation:** Maria Revelj.

**Formal analysis:** Maria Revelj, Karolina Linden.

**Investigation:** Maria Revelj.

**Methodology:** Maria Revelj, Anna Wessberg, Karolina Linden.

**Project administration:** Verena Sengpiel, Karolina Linden.

**Supervision:** Karolina Linden.

**Validation:** Anna Wessberg, Ylva Carlsson, Maria Lindqvist, Verena Sengpiel.

**Writing – original draft:** Maria Revelj, Karolina Linden.

**Writing – review & editing:** Maria Revelj, Anna Wessberg, Ylva Carlsson, Maria Lindqvist, Verena Sengpiel, Karolina Linden.

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
