## [Decision Letter · Decision Letter 0]

24 Mar 2025

Dear Dr. Revelj,

Thank you for submitting your manuscript to PLOS ONE. After careful consideration, we feel that it has merit but does not fully meet PLOS ONE’s publication criteria as it currently stands. Therefore, we invite you to submit a revised version of the manuscript that addresses the points raised during the review process.

**Please submit your revised manuscript in track change mode, a clean version of the revised manuscript and a separate document detailing your responses to the reviewer comments as listed below. **

We look forward to receiving your revised manuscript.

Kind regards,

Paridhi Jha, PhD

Academic Editor

PLOS ONE

**Journal Requirements:**

1. When submitting your revision, we need you to address these additional requirements. Please ensure that your manuscript meets PLOS ONE's style requirements, including those for file naming. The PLOS ONE style templates can be found at https://journals.plos.org/plosone/s/file?id=wjVg/PLOSOne_formatting_sample_main_body.pdf and https://journals.plos.org/plosone/s/file?id=ba62/PLOSOne_formatting_sample_title_authors_affiliations.pdf 2. Please amend either the abstract on the online submission form (via Edit Submission) or the abstract in the manuscript so that they are identical. 3. We note that you have indicated that there are restrictions to data sharing for this study. For studies involving human research participant data or other sensitive data, we encourage authors to share de-identified or anonymized data. However, when data cannot be publicly shared for ethical reasons, we allow authors to make their data sets available upon request. For information on unacceptable data access restrictions, please see http://journals.plos.org/plosone/s/data-availability#loc-unacceptable-data-access-restrictions.  Before we proceed with your manuscript, please address the following prompts: a) If there are ethical or legal restrictions on sharing a de-identified data set, please explain them in detail (e.g., data contain potentially identifying or sensitive patient information, data are owned by a third-party organization, etc.) and who has imposed them (e.g., a Research Ethics Committee or Institutional Review Board, etc.). Please also provide contact information for a data access committee, ethics committee, or other institutional body to which data requests may be sent. b) If there are no restrictions, please upload the minimal anonymized data set necessary to replicate your study findings to a stable, public repository and provide us with the relevant URLs, DOIs, or accession numbers. Please see http://www.bmj.com/content/340/bmj.c181.long for guidelines on how to de-identify and prepare clinical data for publication. For a list of recommended repositories, please see https://journals.plos.org/plosone/s/recommended-repositories. You also have the option of uploading the data as Supporting Information files, but we would recommend depositing data directly to a data repository if possible. Please update your Data Availability statement in the submission form accordingly. 

Reviewers' comments:

Reviewer's Responses to Questions

**Comments to the Author**

1. Is the manuscript technically sound, and do the data support the conclusions?

Reviewer #1: Partly

Reviewer #2: Yes

2. Has the statistical analysis been performed appropriately and rigorously?

Reviewer #1: N/A

Reviewer #2: N/A

3. Have the authors made all data underlying the findings in their manuscript fully available?

Reviewer #1: Yes

Reviewer #2: No

4. Is the manuscript presented in an intelligible fashion and written in standard English?

Reviewer #1: Yes

Reviewer #2: Yes

**Reviewer #1: ** Review Comments on the Manuscript-PONE-D-24-56250

Overall Evaluation

This manuscript presents a compelling and insightful qualitative study on the lived experiences of Swedish women who gave birth while ill with COVID-19. The phenomenological approach effectively captures the depth of participants' emotions and challenges, providing valuable insights for healthcare providers and policymakers. However, there are areas where the study could be strengthened, particularly in methodology, sample justification, thematic analysis, and integration with existing literature.

Strengths of the Manuscript

The study addresses an underexplored area of maternal health during the COVID-19 pandemic.

The use of a phenomenological reflective lifeworld approach allows for an in-depth exploration of lived experiences, contributing to the existing knowledge on maternal healthcare in crises.

The study has clear public health implications, emphasizing the importance of emotional and psychosocial support for women giving birth under extraordinary conditions.

Well-Structured Presentation

The abstract is well-structured and provides a clear summary of the research, including objectives, methodology, key findings, and implications.

The findings are logically presented under well-defined themes ("shame," "distance," and "gratefulness"), making it easy for readers to follow the emotional progression of participants' experiences.

The discussion integrates findings with literature, making important connections between the experiences of these women and existing studies on childbirth, trauma, and the impact of COVID-19.

Ethical Considerations

The study follows rigorous ethical guidelines, securing approval from the Swedish Ethical Review Authority and ensuring informed consent from participants.

The researchers acknowledge potential bias and discuss efforts to "bridle" preconceptions, which enhances the credibility of the qualitative findings.

Weaknesses and Areas for Improvement

Sample Size and Diversity

The study includes only 10 participants, which is relatively small even for qualitative research. While phenomenological studies prioritize depth over breadth, the study would benefit from:

o Justifying why 10 participants were sufficient in capturing data saturation.

Discussing demographic variability (e.g., socio-economic background, education level, healthcare access) to ensure representativeness.

Addressing potential selection bias due to recruitment via hospital Instagram accounts, which may exclude women with limited internet access or digital literacy.

Methodological Rigor

The study claims to use a phenomenological reflective lifeworld approach but does not provide enough detail on how researchers ensured rigor in analysis.

o Were member checks or inter-coder reliability measures used to validate interpretations?

Were any contradictory narratives encountered? If so, how were they reconciled in the thematic analysis?

The use of NVivo for coding is mentioned, but it would be useful to describe how codes were developed and how emergent themes were validated.

Thematic Development

While the three core themes ("shame," "distance," and "gratefulness") are compelling, some nuances may be lost in broad categorization.

For example, "gratefulness" overlaps with the concept of "guilt," as women expressed gratitude despite their fears of burdening healthcare workers.

The theme of "distance" could be further broken down into physical, emotional, and communicative distance, each of which had unique implications for the birthing experience.

Integration with Literature

While the discussion includes previous studies on childbirth and COVID-19, there is limited engagement with broader theories of maternal stress, trauma, and resilience.

Consider applying a conceptual framework (e.g., Lazarus & Folkman’s Cognitive Appraisal Model or Trauma-Informed Care) to contextualize the psychological responses observed in participants.

Comparison with studies from other countries would strengthen the discussion on cultural differences in pandemic maternity care.

Reflexivity and Researcher Influence

The role of the researcher in data collection and interpretation needs to be critically addressed.

The interviewer was not involved in clinical care, but what steps were taken to minimize power imbalances?

Did participants feel they could fully express negative experiences about healthcare settings, given that the researchers were affiliated with medical institutions?

Suggested Revisions

Enhancing Methodological Transparency

Provide a stronger justification for the sample size, explaining how saturation was reached.

Clarify coding procedures and validation strategies (e.g., multiple coders, intercoder reliability).

Discuss whether member-checking was used (e.g., did participants review summaries of their interviews to verify accuracy?).

Expanding Thematic Analysis

Consider refining thematic categories to avoid conceptual overlap:

Instead of "gratefulness," explore the tension between gratitude and guilt.

Instead of "distance," separate physical distance from emotional detachment.

Add counter-narratives or exceptions (e.g., did any women report positive aspects of giving birth while ill?).

Strengthening the Discussion

Incorporate psychological and feminist theories on childbirth, trauma, and resilience.

Compare findings with similar studies from other healthcare settings to provide a global perspective on maternity care during pandemics.

Reflect on how this study contributes to policymaking, particularly in crisis preparedness for maternal healthcare.

Addressing Reflexivity and Ethical Considerations

Discuss the impact of researcher positionality on interview responses.

Explain any steps taken to mitigate power imbalances between the interviewer and participants.

Consider including a limitations section explicitly addressing biases and challenges in data collection.

Final Recommendation

Major Revisions Required

This study makes a valuable contribution to understanding the psychological and emotional impact of giving birth while ill with COVID-19. However, key methodological gaps need to be addressed to strengthen the credibility of findings. By refining thematic analysis, expanding discussion with theoretical frameworks, and improving methodological transparency, the paper can make a stronger impact on maternal healthcare policies and crisis preparedness strategies.

**Reviewer #2:**  Thank you for submitting this insightful manuscript on the lived experiences of Swedish women giving birth while ill with COVID-19. The study contributes valuable qualitative insights into the intersection of childbirth, illness, and healthcare provision during a pandemic. The phenomenological approach is well-justified, and the findings provide a nuanced understanding of the participants' experiences. Below are some suggestions to enhance clarity and conciseness in the manuscript.

1. Redundancy in the Presentation of Participant Characteristics

The table detailing participant characteristics presents information that is already well described in the text. As the textual explanation provides a clearer understanding of the data, I recommend keeping the narrative and reducing the table to only non-redundant information. Alternatively, the table could be reformatted to emphasize key trends rather than listing all variables already covered in the text.

2. Redundancy at the Beginning of the Results Section

The introduction to the results section repeats some information already covered in the methodology and introduction. I suggest streamlining this part to focus directly on the key findings. A more direct transition to the "essence" of the phenomenon could improve readability and avoid repetition.

3. General Clarity and Conciseness

While the phenomenological approach requires a rich narrative, there are sections where certain descriptions could be made more concise without losing depth. For example:

- Some sections reiterate concepts of shame, guilt, and gratitude multiple times. Consolidating these discussions where appropriate could strengthen the impact of the findings.

- The discussion could more explicitly connect the findings to practical recommendations for healthcare professionals, particularly in how to mitigate the emotional and psychological burden experienced by birthing women in crisis contexts.

4. Data Availability Statement

The current data availability statement notes that the data contain sensitive information and cannot be shared openly. While this is understandable, PLOS ONE requires clear justifications for any data-sharing restrictions. It may be helpful to specify if de-identified excerpts or aggregated data could be made available under controlled conditions.

These revisions would enhance the clarity and impact of the manuscript while maintaining the depth of analysis. I appreciate the effort put into this study and believe that with these minor refinements, the manuscript will make a strong contribution to the field.

Best regards,

**Do you want your identity to be public for this peer review?** For information about this choice, including consent withdrawal, please see our Privacy Policy

Reviewer #1: **Yes: ** I am signing it on my behalf. My name to be appear Sadika Akhter

Reviewer #2: **Yes: ** Dr. Carolina Alday-Mondaca

---

## [Author Response · Author response to Decision Letter 1]

28 Apr 2025

Reviewer #1: Review Comments on the Manuscript

Overall Evaluation

This manuscript presents a compelling and insightful qualitative study on the lived experiences of Swedish women who gave birth while ill with COVID-19. The phenomenological approach effectively captures the depth of participants' emotions and challenges, providing valuable insights for healthcare providers and policymakers. However, there are areas where the study could be strengthened, particularly in methodology, sample justification, thematic analysis, and integration with existing literature.

Response: Thank you for your positive feedback regarding our manuscript. We are grateful for your thorough review that is very helpful in improving the reporting of the study.

Strengths of the Manuscript

The study addresses an underexplored area of maternal health during the COVID-19 pandemic.

The use of a phenomenological reflective lifeworld approach allows for an in-depth exploration of lived experiences, contributing to the existing knowledge on maternal healthcare in crises.

The study has clear public health implications, emphasizing the importance of emotional and psychosocial support for women giving birth under extraordinary conditions.

Well-Structured Presentation

The abstract is well-structured and provides a clear summary of the research, including objectives, methodology, key findings, and implications.

The findings are logically presented under well-defined themes ("shame," "distance," and "gratefulness"), making it easy for readers to follow the emotional progression of participants' experiences.

The discussion integrates findings with literature, making important connections between the experiences of these women and existing studies on childbirth, trauma, and the impact of COVID-19.

Ethical Considerations

The study follows rigorous ethical guidelines, securing approval from the Swedish Ethical Review Authority and ensuring informed consent from participants.

The researchers acknowledge potential bias and discuss efforts to "bridle" preconceptions, which enhances the credibility of the qualitative findings.

Response: Thank you for acknowledging the study’s strengths.

Weaknesses and Areas for Improvement

Sample Size and Diversity

The study includes only 10 participants, which is relatively small even for qualitative research. While phenomenological studies prioritize depth over breadth, the study would benefit from:

o Justifying why 10 participants were sufficient in capturing data saturation.

R1: Response: Thank you for your insightful question. You are correct in that phenomenological studies prioritize depth over breadth. Phenomenology according to Dahlberg et al is strongly linked to philosophy and does not use saturation, rather variation and depth of the phenomena. A usual sample size for this analysis is 8-12 participants, with 10 being typical. We have now clarified this in the methods section: The sample size was determined in line with phenomenological methodology, which prioritises depth of experience over breadth1. Please see line number 199-200.

Discussing demographic variability (e.g., socio-economic background, education level, healthcare access) to ensure representativeness.

R2: Response: Thank you we have added the following text to the methods section: Participants represented a variety of obstetric and health-related backgrounds, including both primiparas and multiparas, and women with and without pregnancy complications. Please see line numbers 200-202.

Addressing potential selection bias due to recruitment via hospital Instagram accounts, which may exclude women with limited internet access or digital literacy.

R3: Response: Thank you for this important comment. We have clarified that we recruited participants in two ways in the methods section, please see line 192-193. We have also added a statement regarding this limitation and its potential impact regarding transferability in the discussion: We also acknowledge that the part of the recruitment that took place via the hospital's Instagram account may have inadvertently excluded women with lower levels of digital literacy, which could in turn affect the transferability of the findings. (Please see, line numbers 592-595)

Methodological Rigor

The study claims to use a phenomenological reflective lifeworld approach but does not provide enough detail on how researchers ensured rigor in analysis.

o Were member checks or inter-coder reliability measures used to validate interpretations?

Were any contradictory narratives encountered? If so, how were they reconciled in the thematic analysis?

R4: Response: Thank you. In phenomenology, the researcher interprets rather than describes the data (see Dahlberg H, Dahlberg K. The question of meaning-a momentous issue for qualitative research. Int J Qual Stud Health Well-being. 2019 Dec;14(1):1598723), in pursuit of meaning. Therefore, inter-coder reliability is not measured, nor is member-checking utilised. However, the analysis was continuously discussed with the senior author and checked against the text to make sure that the result was grounded in the data. Further, all authors confirmed the analysis and the essence in discussion. We have tried to describe this process as openly as possible in accordance with the SRQR checklist (see line numbers 618-619 as well as 605-609). Please specify if you have suggestions for details missing and we will gladly provide them.

The use of NVivo for coding is mentioned, but it would be useful to describe how codes were developed and how emergent themes were validated.

R5: Response: Thank you for the question. NVivo was only used to organize the transcripts and find meaning units to form clusters. All analysis was done by the main author in discussion with the last author and no AI was used.

Thematic Development

While the three core themes ("shame," "distance," and "gratefulness") are compelling, some nuances may be lost in broad categorization.

For example, "gratefulness" overlaps with the concept of "guilt," as women expressed gratitude despite their fears of burdening healthcare workers.

The theme of "distance" could be further broken down into physical, emotional, and communicative distance, each of which had unique implications for the birthing experience.

R6: Response: Thank you, we agree that there is an overlap between some of the constituents, this is common in phenomenology and is further described in the essence namely ‘Reclaiming motherhood through shame, distance and gratefulness’. We thank you for pointing out the difference in distance and have added this in the explanation of the essence: Therefore, the distance emerged as physical, emotional and communicative, which all impacted the birth experience. (see line number 279-281)

Integration with Literature

While the discussion includes previous studies on childbirth and COVID-19, there is limited engagement with broader theories of maternal stress, trauma, and resilience.

Consider applying a conceptual framework (e.g., Lazarus & Folkman’s Cognitive Appraisal Model or Trauma-Informed Care) to contextualize the psychological responses observed in participants.

Comparison with studies from other countries would strengthen the discussion on cultural differences in pandemic maternity care.

R7: Response: Thank you for this excellent suggestion that truly adds to our findings. We have added a discussion about trauma-informed care, please see line numbers 572-580. Regarding the comparison with other countries: Unfortunately, there are very few articles writing about the topic, even from an international perspective. We have discussed the ones that we have found.

Reflexivity and Researcher Influence

The role of the researcher in data collection and interpretation needs to be critically addressed.

The interviewer was not involved in clinical care, but what steps were taken to minimize power imbalances?

Did participants feel they could fully express negative experiences about healthcare settings, given that the researchers were affiliated with medical institutions?

R8: Response: Thank you for raising this important question. Several steps were taken to address the power imbalance in the interview setting. All interviews were conducted digitally, meaning that the participant was comfortable in her own home setting. The interviewer presented herself as a PhD-student, not as an obstetrician. She also used her own casual clothing, mostly hoodies or fleece jackets, and the background was neutral with no markings of a hospital setting. Please see line numbers 212-216.

Suggested Revisions

Enhancing Methodological Transparency

Provide a stronger justification for the sample size, explaining how saturation was reached.

Response: please see R1 above.

Clarify coding procedures and validation strategies (e.g., multiple coders, intercoder reliability).

Discuss whether member-checking was used (e.g., did participants review summaries of their interviews to verify accuracy?).

Response: please see R4 above.

Expanding Thematic Analysis

Consider refining thematic categories to avoid conceptual overlap:

Instead of "gratefulness," explore the tension between gratitude and guilt.

Instead of "distance," separate physical distance from emotional detachment.

Add counter-narratives or exceptions (e.g., did any women report positive aspects of giving birth while ill?).

Response: Please see R6 above.

Strengthening the Discussion

Incorporate psychological and feminist theories on childbirth, trauma, and resilience.

Compare findings with similar studies from other healthcare settings to provide a global perspective on maternity care during pandemics.

Response: Please see R7 above.

Reflect on how this study contributes to policymaking, particularly in crisis preparedness for maternal healthcare.

R9: Response: Thank you for addressing the need to further discuss contributions to policymaking, we have added this to the study’s conclusion. This study provides valuable insights for policymaking in strengthening crisis preparedness within maternal healthcare. Our findings highlight the need for policies that not only ensure physical safety and infection control but also protect women’s dignity, and emotional well-being during childbirth. In particular, the study underscores the importance of maintaining psychosocial support and respectful maternity care, even in the context of public health emergencies. Please see line numbers 633-636.

Addressing Reflexivity and Ethical Considerations

Discuss the impact of researcher positionality on interview responses.

Explain any steps taken to mitigate power imbalances between the interviewer and participants.

Consider including a limitations section explicitly addressing biases and challenges in data collection. Addressing Reflexivity and Ethical Considerations. Explain any steps taken to mitigate power imbalances between the interviewer and participants.

Response: Please see R8 above.

Final Recommendation

Major Revisions Required

This study makes a valuable contribution to understanding the psychological and emotional impact of giving birth while ill with COVID-19. However, key methodological gaps need to be addressed to strengthen the credibility of findings. By refining thematic analysis, expanding discussion with theoretical frameworks, and improving methodological transparency, the paper can make a stronger impact on maternal healthcare policies and crisis preparedness strategies.

Reviewer #2: Thank you for submitting this insightful manuscript on the lived experiences of Swedish women giving birth while ill with COVID-19. The study contributes valuable qualitative insights into the intersection of childbirth, illness, and healthcare provision during a pandemic. The phenomenological approach is well-justified, and the findings provide a nuanced understanding of the participants' experiences. Below are some suggestions to enhance clarity and conciseness in the manuscript.

Response: Thank you for your positive feedback regarding our manuscript. We are grateful for your thorough review that is very helpful in improving the reporting of the study.

1. Redundancy in the Presentation of Participant Characteristics

The table detailing participant characteristics presents information that is already well described in the text. As the textual explanation provides a clearer understanding of the data, I recommend keeping the narrative and reducing the table to only non-redundant information. Alternatively, the table could be reformatted to emphasize key trends rather than listing all variables already covered in the text.

Response: Thank you, we agree that there was repetition. We have omitted the table all together since it did not convey any further information to the reader. Please see line number 246-259.

2. Redundancy at the Beginning of the Results Section

The introduction to the results section repeats some information already covered in the methodology and introduction. I suggest streamlining this part to focus directly on the key findings. A more direct transition to the "essence" of the phenomenon could improve readability and avoid repetition.

Response: Thank you. We have omitted the definitions of mild, moderate and severe covid to avoid repetition. The description of the essence of the phenomena needs to be deep and a few concepts of repletion of factors from the background are to be expected. Yet, we have tried to enhance clarity in the text.

3. General Clarity and Conciseness

While the phenomenological approach requires a rich narrative, there are sections where certain descriptions could be made more concise without losing depth. For example:

- Some sections reiterate concepts of shame, guilt, and gratitude multiple times. Consolidating these discussions where appropriate could strengthen the impact of the findings.

Response: Thank you. We have tried to make the text more concise without losing meaning. If you have further concrete suggestions, it would be much appreciated.

- The discussion could more explicitly connect the findings to practical recommendations for healthcare professionals, particularly in how to mitigate the emotional and psychological burden experienced by birthing women in crisis contexts.

Response: Thank you for this important point! We have added a discussion about trauma informed care and made sure to include care interventions to make it more useful for healthcare professionals. Please see response R7 to reviewer 1 above as well as line number 572-580.

4. Data Availability Statement

The current data availability statement notes that the data contain sensitive information and cannot be shared openly. While this is understandable, PLOS ONE requires clear justifications for any data-sharing restrictions. It may be helpful to specify if de-identified excerpts or aggregated data could be made available under controlled conditions.

Response: Thank you. We agree with the commitment to open science. Regrettably the Swedish ethical review authority has yet to provide guidelines regarding how qualitative data is to be shared openly. This strongly limits our possibility to share data openly.

Data Availability Statement: Data contains potentially identifying and sensitive information, and there are ethical restrictions to sharing the data file as it includes individual interviews with participants. For data requests, please contact The Swedish Ethical Review Authority: registrator@etikprovning.se. Any specific requests concerning de-identified excerpts or aggregated data can be made directly to the corresponding author pending approval by the University of Gothenburg. Line numbers 640-646.

These revisions would enhance the clarity and impact of the manuscript while maintaining the depth of analysis. I appreciate the effort put into this study and believe that with these minor refinements, the manuscript will make a strong contribution to the field.

1. Dahlberg K, Dahlberg H, Nyström M. Reflective Lifeworld Research. second edition ed. Lund: Studentlitteratur AB; 2008.

---

## [Decision Letter · Decision Letter 1]

1 Sep 2025

Dear Dr. Revelj,

Thank you for submitting your manuscript to PLOS ONE. After careful consideration, we feel that it has merit but does not fully meet PLOS ONE’s publication criteria as it currently stands. Therefore, we invite you to submit a revised version of the manuscript that addresses the points raised during the review process.

We look forward to receiving your revised manuscript.

Kind regards,

Paridhi Jha, PhD

Academic Editor

PLOS ONE

**Journal Requirements:**

Reviewers' comments:

Reviewer's Responses to Questions

**Comments to the Author**

Reviewer #3: (No Response)

Reviewer #4: (No Response)

2. Is the manuscript technically sound, and do the data support the conclusions?

Reviewer #3: Yes

Reviewer #4: Yes

3. Has the statistical analysis been performed appropriately and rigorously?

Reviewer #3: N/A

Reviewer #4: N/A

4. Have the authors made all data underlying the findings in their manuscript fully available?

Reviewer #3: Yes

Reviewer #4: Yes

5. Is the manuscript presented in an intelligible fashion and written in standard English?

Reviewer #3: Yes

Reviewer #4: Yes

**Reviewer #3: ** Reclaiming Motherhood through Shame, Distance, and Gratefulness - A Phenomenological Study of Swedish Women’s Lived experiences of giving birth while Ill with COVID-19

The primary aim of this study is to apply a phenomenological approach to explore the experiences of women who gave birth while infected with SARS-CoV-2 in Sweden. The study offers valuable insights from both health systems and policy perspectives. As social and healthcare practitioners are often required to navigate complex challenges in their work environments, the findings contribute meaningfully to our understanding of real-world practice. The study has undergone reviews. It does meet the requisite ethical and research standards. Some questions that arose are if the authors reflected on ‘bracketing or epoché’ though difficult to implement in practical sense. As traditional reliability and validity tools are not relevant in a phenomenological study. What were the team processes which enabled reflexivity into data-analysis? If the authors add a few more details into the data analysis, it would be useful to scholars of phenomenology approach.

Which language were the interviews conducted in? How did translation impact the data analysis? Thank you,

**Reviewer #4:**  This is a well-conceived and timely phenomenological study exploring women’s lived experiences of giving birth while ill with COVID-19 in Sweden. The revisions since the first review have strengthened the manuscript considerably, the methodology is now clearer, thematic analysis more nuanced, reflexivity better articulated, and the conclusion more firmly linked to crisis preparedness. The paper makes an important contribution to maternal health research in crisis contexts. With a few further refinements, the manuscript can achieve greater methodological transparency, conceptual precision, and integration with the broader qualitative literature.

Major Points

1. Trustworthiness and Rigor

While the reflective lifeworld approach and bridling are well explained, it would strengthen the manuscript to include a short paragraph mapping the study procedures explicitly to trustworthiness criteria (credibility, dependability, confirmability, transferability). For example: iterative immersion, analyst triangulation, and grounding in quotations (credibility); audit trail and reflexive discussions (dependability/confirmability); thick description of context and participant variation (transferability). Referencing Lincoln & Guba or Creswell’s validation strategies would enhance methodological transparency for readers less familiar with phenomenology.

2. Thematic Precision

The subdivision of distance into physical, emotional, and communicative is excellent. However, gratefulness remains conceptually entangled with guilt and shame. Consider renaming this theme ambivalent gratitude or explicitly discussing the intertwining of gratitude and guilt in the Results section, to avoid interpretive ambiguity. The consistent use of gratitude rather than gratefulness is recommended.

3. Integration with Broader Theory

The addition of trauma-informed care and stigma literature is valuable. To situate the findings more firmly, a short paragraph in the Discussion linking to Lazarus & Folkman’s stress–coping framework (appraisal, coping strategies) would be helpful, as would a brief reference to feminist childbirth literature (medicalization, agency, relational power). These connections would show that the insights extend beyond the pandemic context.

4. Reflexivity and Positionality

The reflexivity section is notably improved (non-clinical stance, casual clothing, digital interviews). To further strengthen it, one line acknowledging the unavoidable positionality associated with institutional affiliation in obstetric/midwifery contexts would be useful, along with a brief note on how this was mitigated (e.g., emphasizing confidentiality, explicitly inviting critique).

5. Policy and Practice Implications

The conclusion usefully links the findings to maternal healthcare preparedness in crises. To maximize the practical relevance, consider integrating implications more directly into the Discussion by including short practice pointers aligned with each theme (e.g., humanizing PPE, providing emotional reassurance, establishing clear protocols for early skin-to-skin contact, ensuring psychological follow-up for severe maternal illness).

Minor Points

- Abstract: revise phrasing to reclaiming motherhood while navigating shame, distance, and gratitude.

- Quotations: some excerpts are overly long (over eight lines). Please trim the longest to maintain analytic focus while preserving participants’ voices.

- Language/consistency: correct “it’s constituents” → “its constituents”; standardize % formatting (e.g., 80%); tidy hyphenation (post-birth, COVID-19 related).

- References: coverage is strong, but a final formatting pass for capitalization and URLs is recommended.

(Q4) Data availability

Although full transcripts cannot be shared due to ethical restrictions, deidentified quotations are included in the manuscript and additional excerpts are available under ethics approval. This is consistent with PLOS ONE’s data policy for qualitative studies.

(Q5) English presentation

The manuscript is clear and written in standard English. Only minor copyediting is required for consistency and readability.

**Do you want your identity to be public for this peer review?** For information about this choice, including consent withdrawal, please see our Privacy Policy

Reviewer #3: **Yes: ** Keerty Nakray

Reviewer #4: No

---

## [Author Response · Author response to Decision Letter 2]

17 Sep 2025

Reviewer #3: Review Comments on the Manuscript

Reclaiming Motherhood through Shame, Distance, and Gratefulness - A Phenomenological Study of Swedish Women’s Lived experiences of giving birth while Ill with COVID-19

The primary aim of this study is to apply a phenomenological approach to explore the experiences of women who gave birth while infected with SARS-CoV-2 in Sweden. The study offers valuable insights from both health systems and policy perspectives. As social and healthcare practitioners are often required to navigate complex challenges in their work environments, the findings contribute meaningfully to our understanding of real-world practice. The study has undergone reviews. It does meet the requisite ethical and research standards. Some questions that arose are if the authors reflected on ‘bracketing or epoché’ though difficult to implement in practical sense. As traditional reliability and validity tools are not relevant in a phenomenological study. What were the team processes which enabled reflexivity into data-analysis? If the authors add a few more details into the data analysis, it would be useful to scholars of phenomenology approach.

Which language were the interviews conducted in? How did translation impact the data analysis? Thank you

Response: Thank you very much for your kind and encouraging feedback on our manuscript. We truly appreciate the time and care you devoted to reviewing our work. Your thoughtful comments have been very helpful in strengthening the clarity and reporting of the study, and we are sincerely grateful for your support.

In response to your questions, we have carefully revised the manuscript and addressed each point raised. Where appropriate, we have incorporated new clarifications into the manuscript and provided line references below.

What were the team processes which enabled reflexivity into data-analysis? If the authors add a few more details into the data analysis, it would be useful to scholars of phenomenology approach.

Thank you for this suggestion. We have added further details. Please see line number 147-151, 235-239, 641-647.

Which language were the interviews conducted in? How did translation impact the data analysis?

Thank you, this is an important point. All interviews were conducted in Swedish. Please see line number 207. We have now also included an explanation of our approach to translating the interview quotations. Please see line number 652-655.

Reviewer #4: This is a well-conceived and timely phenomenological study exploring women’s lived experiences of giving birth while ill with COVID-19 in Sweden. The revisions since the first review have strengthened the manuscript considerably, the methodology is now clearer, thematic analysis more nuanced, reflexivity better articulated, and the conclusion more firmly linked to crisis preparedness. The paper makes an important contribution to maternal health research in crisis contexts. With a few further refinements, the manuscript can achieve greater methodological transparency, conceptual precision, and integration with the broader qualitative literature.

Response: We sincerely thank you for your constructive, thoughtful, and generous feedback. We greatly appreciate the time and care invested in reviewing our manuscript. The comments have helped us to significantly enhance the methodological clarity, conceptual rigour, and practical relevance of the study.

In response to the suggestions, we have carefully revised the manuscript and addressed each point raised. Where appropriate, we have incorporated new clarifications into the manuscript and provided line references below.

Please find our point-by-point responses to each comment below.

Major Points

1. Trustworthiness and Rigor

While the reflective lifeworld approach and bridling are well explained, it would strengthen the manuscript to include a short paragraph mapping the study procedures explicitly to trustworthiness criteria (credibility, dependability, confirmability, transferability). For example: iterative immersion, analyst triangulation, and grounding in quotations (credibility); audit trail and reflexive discussions (dependability/confirmability); thick description of context and participant variation (transferability). Referencing Lincoln & Guba or Creswell’s validation strategies would enhance methodological transparency for readers less familiar with phenomenology.

Response: We are grateful for the reviewer’s thoughtful suggestion regarding Creswell’s validation strategies. While we recognize their importance in many qualitative traditions, we believe that directly applying these strategies would not be fully appropriate in the present study, as they are not grounded in phenomenological philosophy and may, in some respects, be at odds with its methodological commitments. Instead, we have followed the quality criteria articulated by Dahlberg, Dahlberg, and Nyström (2008), where rigour is ensured through openness, bridling of pre-understandings, and a systematic search for the essence and variations of lived experience. Phenomenology seeks to understand and describe the subjective experience of a phenomenon. To strengthen trustworthiness, we sought to describe the women’s lived experiences with openness, nuance, and methodological transparency. With respect to credibility, we have presented rich empirical excerpts to illustrate our interpretations. For confirmability, the identified essence and constituents were critically discussed and validated with co-authors who were not involved in the primary analysis. Regarding dependability, we ensured transparency by carefully documenting the analytic process, moving iteratively between whole and parts, and by engaging in reflexive discussions about our pre-understandings throughout the study.

We have incorporated parts of the above text into the discussion of the methodology section. We have also addressed the remaining points and integrated relevant reflections where they aligned with the structure and content of the manuscript.

Please see line number 606-614.

2. Thematic Precision

The subdivision of distance into physical, emotional, and communicative is excellent. However, gratefulness remains conceptually entangled with guilt and shame. Consider renaming this theme ambivalent gratitude or explicitly discussing the intertwining of gratitude and guilt in the Results section, to avoid interpretive ambiguity. The consistent use of gratitude rather than gratefulness is recommended.

Response: We sincerely thank the reviewer for this insightful observation. In response, we have revised the manuscript to reflect the suggested phrasing and now use gratitude consistently throughout the manuscript instead of gratefulness, to enhance conceptual clarity.

3. Integration with Broader Theory

The addition of trauma-informed care and stigma literature is valuable. To situate the findings more firmly, a short paragraph in the Discussion linking to Lazarus & Folkman’s stress–coping framework (appraisal, coping strategies) would be helpful, as would a brief reference to feminist childbirth literature (medicalization, agency, relational power). These connections would show that the insights extend beyond the pandemic context.

Response: We sincerely thank the reviewer for this thoughtful suggestion. We agree that Lazarus and Folkman’s stress–coping framework offers an important perspective for understanding how individuals appraise and respond to challenging situations. However, since our study is firmly situated within a phenomenological lifeworld approach, we have chosen to keep the discussion within this methodological and theoretical tradition to avoid diluting the phenomenological focus. We therefore respectfully refrain from adding this additional framework, but we greatly appreciate the insight and acknowledge its relevance for readers who may wish to interpret our findings through complementary theoretical lenses.

4. Reflexivity and Positionality

The reflexivity section is notably improved (non-clinical stance, casual clothing, digital interviews). To further strengthen it, one line acknowledging the unavoidable positionality associated with institutional affiliation in obstetric/midwifery contexts would be useful, along with a brief note on how this was mitigated (e.g., emphasizing confidentiality, explicitly inviting critique).

Response: Thank you for this thoughtful and constructive suggestion. We fully agree that acknowledging the unavoidable positionality linked to institutional affiliation is important in enhancing reflexivity and transparency. We have now added a sentence in the reflexivity section acknowledging the first author’s (MR) institutional affiliation in obstetrics and clarifying steps taken to mitigate this, including emphasising confidentiality, voluntary participation, and a bridled, open stance in interviews.

Please see line numbers 641 – 647

5. Policy and Practice Implications

The conclusion usefully links the findings to maternal healthcare preparedness in crises. To maximize the practical relevance, consider integrating implications more directly into the Discussion by including short practice pointers aligned with each theme (e.g., humanizing PPE, providing emotional reassurance, establishing clear protocols for early skin-to-skin contact, ensuring psychological follow-up for severe maternal illness).

Response: Thank you for this valuable and constructive suggestion. We have now integrated specific practice-oriented reflections into the Discussion section, in alignment with each theme. These additions include insights such as the importance of humanising care even when PPE is required, the therapeutic value of respectful presence, the significance of facilitating early skin-to-skin contact when feasible, and the need for structured psychological follow-up for severely ill mothers. We believe these refinements strengthen the study’s relevance for crisis preparedness in maternal healthcare

Please see line number: 541-543, 559-562, 572-574, 594-596.

Minor Points

- Abstract: revise phrasing to reclaiming motherhood while navigating shame, distance, and gratitude.

- Quotations: some excerpts are overly long (over eight lines). Please trim the longest to maintain analytic focus while preserving participants’ voices.

- Language/consistency: correct “it’s constituents” → “its constituents”; standardize % formatting (e.g., 80%); tidy hyphenation (post-birth, COVID-19 related).

- References: coverage is strong, but a final formatting pass for capitalization and URLs is recommended.

Response: Thank you for these helpful suggestions. In response, we have revised the abstract to reflect the suggested phrasing and now use gratitude consistently throughout the manuscript instead of gratefulness, to enhance conceptual clarity. We have reviewed the quotations and shortened those that exceeded eight lines, ensuring the analytic focus is preserved while retaining participants’ voices. Additionally, we have carefully edited the manuscript for improved language clarity and consistency, including correction of typographical errors (e.g., “it’s” to “its). A final review of the reference list was also conducted to ensure accuracy and formatting consistency.

---

## [Editor Report · Decision Letter 2]

22 Sep 2025

Subject: Submission of original Research article

Reclaiming Motherhood through Shame, Distance, and Gratitude - A Phenomenological Study of Swedish Women’s Lived experiences of giving birth while Ill with COVID-19

PONE-D-24-56250R2

Dear Dr. Revelj,

We’re pleased to inform you that your manuscript has been judged scientifically suitable for publication and will be formally accepted for publication once it meets all outstanding technical requirements.

Kind regards,

Paridhi Jha, PhD

Academic Editor

PLOS ONE
---

## [Editor Report · Acceptance letter]

PONE-D-24-56250R2

PLOS ONE

Dear Dr. Revelj,

I'm pleased to inform you that your manuscript has been deemed suitable for publication in PLOS ONE. Congratulations! Your manuscript is now being handed over to our production team.

Kind regards,

on behalf of

Dr. Paridhi Jha

Academic Editor

PLOS ONE